# The Mental Health Outcomes and Cost Estimates of Korean Medicine for Anxiety Disorder Patients

**DOI:** 10.3390/healthcare12131345

**Published:** 2024-07-05

**Authors:** So-Young Lee, Jong-Ho Yoo, Sang-Il Seo, Ji-Eun Lee, Geun-Woo Kim, Eun Cho

**Affiliations:** 1College of Pharmacy, Sookmyung Women’s University, Cheongpa-ro 47-gil 100, Yongsan-gu, Seoul 04310, Republic of Korea; soyeong511@gmail.com (S.-Y.L.); jieun1608@naver.com (J.-E.L.); 2Haneum Neuropsychiatry Clinic of Korean Medicine, 37, Eonju-ro 98-gil, Gangnam-gu, Seoul 06148, Republic of Korea; fcodefree@naver.com; 3Haneum Neuropsychiatry Clinic of Korean Medicine, 118, Sangnam-ro, Seongsan-gu, Changwon-si 51495, Gyeongsangnam-do, Republic of Korea; tjtkddlf4458@naver.com; 4Department of Neuropsychiatry, Dongguk University Bundang Oriental Hospital, 268 Buljeong-ro Bundang-gu, Seongnam-si 13601, Gyeonggi-do, Republic of Korea

**Keywords:** anxiety disorder, cost analysis, Korean medicine, quality of life, retrospective study

## Abstract

Korean medicine (KM) is used to treat anxiety disorders, but there is limited research on its effects. This study aimed to examine the associations between improved QoL and reduced clinical symptoms and KM in patients with anxiety disorders. The medical records of patients with anxiety who were treated with KM (acupuncture, psychotherapy, Chuna therapy, aromatherapy, or herbal medicine) for at least 4 weeks were retrospectively analyzed. Clinical, QoL, and cost outcomes were measured at baseline and at weeks 4 and 12 (Anxiety: State-Trait Anxiety Inventory [STAI X-1 (state), X-2 (trait)], Beck Anxiety Inventory [BAI]; anger: State-Trait Anger Expression Inventory State [STAXI-S (state), T (trait)], Anger Expression Inventory [AXI-K-I (anger-in), AXI-K-O (anger-out), AXI-K-C (anger-control); depression: Beck Depression Inventory-II [BDI II], QoL: QoL-related instruments Euro Quality of Life 5 Dimensions utility score [EQ-5D], Euro QoL Visual Analog Scale [EQ-VAS]). The total costs for each item were calculated in terms of NHIS-covered costs and patients’ out-of-pocket costs from the perspective of the healthcare system. The medical records of 67 patients were evaluated. The KM treatments were found to be associated with decreased anxiety (STAI X-1; STAI X-2; BAI, *p* < 0.0001), depression (BDI-II, *p* < 0.0001), and anger (AKI-K-I; AKI-K-O, *p* < 0.05) and increased QoL (EQ-5D; EQ-VAS, *p* < 0.0001). An average of USD 1360 was paid for the KM treatments for 4 weeks. The study findings suggested that KM may improve clinical symptoms and QoL outcomes in patients with anxiety disorders.

## 1. Introduction

Anxiety is defined as a ‘persistent feeling of dread, apprehension, and impending disaster or tension and uneasiness’ [1]. Anxiety disorders are the most common type of mental disorder and include generalized anxiety disorder (GAD), panic disorder (PD), phobias, obsessive-compulsive disorder, and post-traumatic stress disorder (PTSD) due to general medical conditions [2]. These disorders are associated with considerable distress, impaired functioning, and increased risk of suicide [3]. Over 301 million people worldwide, representing approximately 3.9% of the world’s population, were estimated to be affected by anxiety disorders, accounting for the largest proportion of all mental diseases (31.1%) in 2019 [4,5]. Women are known to be 1.5 to 2 times more likely to be diagnosed with anxiety disorders than men [2]. In Korea, the lifetime prevalence rate of anxiety disorders was 9.3%, and the annual prevalence rate was 3.1% in 2021, which have both been increasing each year [6].

While various pharmacological and psychological therapies exist for anxiety disorders, patients with these conditions often opt for the use of complementary and alternative medicine (CAM) to decrease or avoid medication use, alleviate physical and psychological symptoms, and improve quality of life (QoL) [7]. Moreover, CAM can help with areas of mental health that are difficult to address using Western medicine (WM) [8,9]. In Korea, a unique dual medical system exists in which both WM and Korean medicine (KM) are available to patients with anxiety disorders [10]. KM uses a symptom-based approach that identifies specific etiologies and symptoms rather than prioritizing the naming of specific diseases or diagnoses, and it utilizes various treatments such as herbal medicines, acupuncture, moxibustion, and physical therapies [11]. These categories are specific to KM and may not directly correspond to the classifications of CAM therapies used in other countries [9,12,13,14]. While psychotropic drugs are widely used to treat anxiety disorders, patients often seek KM to reduce their dependence on such medications to replace pharmacotherapy altogether and avoid adverse drug reactions [7].

In KM clinics, anxiety is typically diagnosed through self-evaluation, counseling, and anxiety questionnaires, such as the State-Trait Anxiety Inventory (STAI) and Beck Anxiety Inventory (BAI), as well as instrumental examinations, such as heart rate variability (HRV) measurement and neurofeedback [7]. The goal of anxiety treatments in KM is not only to stabilize the mind and alleviate symptoms but also to regulate body function and maintain internal energy balance [15]. In order to achieve this, acupuncture and herbal medicine are commonly used following the treatment principles of regulating the balance between yin and yang, as well as regulating the states of Qi and blood [15]. Psychotherapy, moxibustion, cupping therapy, electroacupuncture, and other techniques may also be used [7,15]. Furthermore, anxiety, depression, and anger are inter-related not only in their co-occurrence but also in their underlying mechanisms and treatment approaches [16,17,18,19]. When measuring the effectiveness of anxiety treatments, multiple dimensions, such as depression, anger, and QoL, are often examined along with anxiety symptoms [16,17,20]. In addition, previous studies on KM therapy have only examined the effectiveness of specific KM treatments for anxiety disorders [21,22], so there is no research on the impact of KM treatment regimens for anxiety disorders. Thus, it is necessary to examine how the symptoms of anger or depression improve when anxiety symptoms get better with KM treatments in patients with anxiety.

The present study aimed to investigate the impact of KM treatment regimens (acupuncture, KM psychotherapy, Chuna therapy, aromatherapy, and herbal medicine) on patients with anxiety disorders. We examined medical records from KM clinics to evaluate clinical outcomes, QoL, and improvement in other symptoms related to KM treatment. We also estimated the costs related to treatments being employed for patients with anxiety in the KM clinic.

## 2. Materials and Methods

### 2.1. Study Design

A retrospective observational study design was adopted to evaluate the clinical outcomes, QoL outcomes, and costs of KM treatments. We reviewed the medical records of patients with anxiety who visited one of two neuropsychiatric Korean medicine (NKM) clinics in Seoul. The study was carried out in accordance with the Code of Ethics of the World Medical Association; however, informed consent was not required due to the retrospective nature of the study.

### 2.2. Study Population

The study population included patients who first visited one of the two NKM clinics between 1 December 2017 and 30 June 2020. All patients were aged 20 years or older upon their first visit and had been diagnosed with social anxiety disorder (SAD), PD, PTSD, or GAD according to the criteria of the Diagnostic and Statistical Manual of Mental Disorders, Fifth Edition (DSM-5). The demographic data, clinical and QoL outcomes, and expenditures for each treatment of those patients who received treatments for more than 4 weeks were extracted from the medical records to ensure the detection of minimum significant changes.

### 2.3. Korean Medicine Intervention for Anxiety

The diagnostic process in KM involves personality tests, psychiatric history-taking, and HRV measurements upon initial examination by a KM physician. Among the types of treatment in the KM regimen for anxiety disorders, acupuncture, KM psychotherapy, family therapy, aromatherapy, and Chuna therapy were provided weekly. All treatments were covered by the National Health Insurance Service (NHIS), except for herbal medicine and aromatherapy, which needed to be fully paid for by the patients. Herbal medicine, such as Ondamtang or Gwibitang, was usually prescribed for a 1-month supply with daily administration and provided to all patients once a diagnosis was made based on the test results. From weeks 5 to 12, the previous treatment methods other than herbal medicine were provided weekly to patients who visited the clinic; for those requiring continuous herbal treatment, additional herbal medicine was administered. Outcome measurements were conducted at weeks 1, 4, and 12.

### 2.4. Outcome Measurement Variables

In the NKM clinics, patients with anxiety were assessed at treatment initiation and at check-up points, typically occurring at weeks 4 and 12 after treatment initiation. Five domains of anxiety-related scales were examined: (1) Anxiety: STAI axis1 (State, STAI X-1), STAI axis2 (Trait, STAI X-2) [23], and BAI [24]; (2) Anger: State-Trait Anger Expression Inventory State Anger (STAXI-S), State-Trait Anger Expression Inventory Trait Anger (STAXI-T) [25], Anger Expression Inventory Anger-In (AXI-K-I), Anger Expression Inventory Anger-Out (AXI-K-O), and Anger Expression Inventory Anger-Control (AXI-K-C) [26]; (3) Depression: Beck Depression Inventory-II (BDI II) [27]; (4) Optimism: The revised Life Orientation Test (LOT-R) [28]; and (5) Satisfaction: The Satisfaction with Life Scale (SWLS), The Life Satisfaction Expectancy Scale (LSES), and The Life Satisfaction Motivation Scale (LSMS) [29]. A detailed description of the above is provided in Appendix A. In addition, we employed the two QoL-related instruments Euro Quality of Life 5 Dimensions utility score (EQ-5D) and Euro QoL Visual Analog Scale (EQ-VAS). The score ranges and clinical implications are shown in Figure 1. The instruments were completed by the surveyed patients and by a clinical psychologist—who was familiar with the patients and their disorders—at the NKM clinics. The medical costs incurred for the treatments of anxiety disorders in KM clinics can be divided into diagnostic and treatment categories. The diagnostic cost represents a one-time expense incurred during the initial visit, while each item in the treatment cost reflects an average expenditure over a period of 4 or 8 weeks. The total costs for each item were calculated from a health system perspective based on the NHIS-covered costs and patients' out-of-pocket costs. The costs were adjusted to the South Korean won (KRW) in 2020 using the healthcare component of the customer price index from the Korean Statistical Information Service, and the exchange rate from USD to KRW in 2020 (1 USD = KRW 1086.3) was used.

### 2.5. Statistical Analysis

A descriptive analysis of patients’ baseline clinical characteristics was performed. The continuous variables are presented as the mean ± standard deviation (SD) after testing for a normal distribution. The categorical variables are expressed as the number of patients and percentages. The paired *t*-test was used to evaluate changes between the baseline and after 4 weeks of treatment. In order to visualize the underlying relational structures among outcome measurement scales, we utilized a multidimensional scaling technique [30]. A distance matrix was computed using Euclidean distance, and k-means clustering was performed. A correlation analysis was then conducted to determine the strength of the relationship between the scales. Analysis of variance (ANOVA) was conducted to compare the mean differences in outcome measures (STAIX1, STAIX2, EQ-5D, and EQ-VAS) among the combination treatment groups. In all cases, the statistical significance was set at *p* < 0.05. All analyses were performed using R 4.0.2.

## 3. Results

### 3.1. Demographic Characteristics and Clinical Outcomes in Patients

Out of the 106 patients who visited the NKM clinics during the observation period, 67 who were treated for 4 weeks or longer were considered for analysis. The flow chart for patient selection is shown in Figure 2. The mean age of the included patients at their first visit was 33.2 ± 10.7 years, and 42 (62.7%) were female patients. Patients with GAD accounted for 44.8% (n = 30), followed by those with PD (33.8%, n = 26), SAD (13.4%, n = 9), and PTSD (3%, n = 2) (Table 1). The male and female patients did not significantly differ in age, clinical scores, or QoL-related scores, except for the BDI II and EQ-5D scores.

Table 2 shows the changes in clinical outcomes between the baseline and week 4 of treatment. Anxiety levels at the baseline were severe when considering the STAI X-1 (59.24 ± 9.55) and BAI (28.15 ± 12.74) scores or moderate according to the STAI X-2 (57.33 ± 9.98; refer to Figure 1 for the scores and their clinical implications). The patients were found to have severe depression (BDI II, 25.78 ± 12.04) at baseline, whereas the anger-related scores were relatively low. The LOT-R scores were at a low level (11.81 ± 4.04), indicating a low level of optimism, expectancy, or outlook on life. Regarding life satisfaction, at baseline, the participants showed a high level of motivation toward achieving life satisfaction (LSMS, 30.24 ± 3.97), while the level of satisfaction regarding their future life expectations was moderate (LSES, 21.63 ± 7.15). However, the level of satisfaction with their current situation was low (SWLS, 15.31 ± 5.89). The QoL scores were close to optimal, with an EQ-VAS score of approximately 50 and an EQ-5D score of approximately 0.75.

Most of the six anxiety-related domains showed positive improvements after 4 weeks of treatment, with the exception of the factor of anger control (AXI-K-C) in the anger dimension. The greatest statistically significant improvement was found for anxiety levels at 22.5, 14.6, and 9.6% in terms of the BAI, STAI X-1, and STAI X-2 scores, respectively, from the baseline. Further improvements were observed for depression (BDI II, 11.4%, *p* < 0.001), life satisfaction (LSES, 6.2%, *p* = 0.003), and QoL (EQ-5D, 8.4%, *p* < 0.001; EQ-VAS, 11.9%, *p* < 0.001). Among the five anger-related dimensions, both internal (5.0%) and external (3.0%) anger expression showed significant improvement (*p* = 0.012 and 0.047, respectively).

Improvements in clinical outcomes were found in both men and women with similar patterns (Appendix A). Regarding anger, an improvement of 5.5% (*p* = 0.022) was observed in how men expressed their anger (AXI-K-O), whereas women showed a significant enhancement in managing their suppressed anger (AXI-K-I), with a 5.6% increase (*p* = 0.046). The results also showed that men had a statistically significant improvement in the LSES score (7.9%, *p* = 0.019; Appendix A). Four different treatment combinations were evaluated for their effects on improving the symptoms associated with anxiety disorders, including anxiety and quality of life (Appendix A). Herbal medicine, acupuncture, psychotherapy, and aromatherapy were included in all combinations, with the combination receiving all four therapies being the most common (32.8%). The treatment combination that included family therapy showed the most improvement in anxiety symptoms and quality of life (EQ-VAS). However, no significant differences were observed among the four treatment combinations.

### 3.2. Relationships between the Anxiety-Relevant Measurement Scales

Appendix A presents the inter-relationship values among the outcome measurement scales for anxiety disorders. The anxiety indicators demonstrated greater correlations (ρ = 0.43–0.75) with one another and strong relationships with anger and depression indicators, such as STAXI-T (ρ = 0.51–0.56) and BDI II (ρ = 0.44–0.57). The QoL indicators (EQ-5D and EQ-VAS) had a low or moderate correlation with the anxiety parameters. For example, the EQ-VAS scores had the strongest correlations with STAI X-I (ρ = −0.29) and BAI (ρ = −0.22) scores for anxiety and STAXI-S (ρ = −0.23) scores for anger among all examined parameters. The QoL variables showed a greater level of similarity with the measures of anger expression (AXI-K-O) and anger control (AXI-K-C), according to the results of the multidimensional scaling analysis (Appendix A). Likewise, the life satisfaction parameters (SWLS, LSES, and LSMS) were similar to those of optimism (LOT-R; Appendix A).

### 3.3. Costs of Traditional Korean Medicine Treatment for Patients with Anxiety

During the 4 weeks of treatment, the patients visited the KM clinics approximately five times (mean, 5.1; range, 2–9). When the observation period was expanded to 12 weeks (36 patients), the mean number of visits was 11.2 (range, 4–22). The estimated average cost for each item is shown in Appendix A. For 4 weeks, the medical costs amounted to KRW 1,790,222 (USD 1360). Over the subsequent 8 weeks, the costs decreased to KRW 521,108 (USD 396), representing a reduction of approximately 29% compared to the initial four-week period, as the relatively expensive herbal decoction prescription was only given to patients who required it after the 4 weeks of treatment (Figure 3).

## 4. Discussion

Our results demonstrate that the KM approach is effective at enhancing clinical outcomes and QoL in patients with anxiety disorders in real-world clinical settings. In our analysis of the results of the effects of KM (acupuncture, KM psychotherapy, Chuna therapy, aromatherapy, and herbal medicine) for over 4 weeks, there were improvements not only in anxiety but also in depression and anger, along with an increase in the QoL. The medical costs for treating anxiety disorder patients with KM for 4 weeks amounted to KRW 1,790,222 (USD 1,360).

Among previous studies, only two previous randomized controlled trials (RCTs) have shown that KM modalities, such as acupuncture and herbal medicine, had a positive impact on both the clinical outcomes and the QoL in patients with anxiety [21,22]. A meta-analysis that included 20 RCTs showed that acupuncture was more effective than conventional treatments in reducing anxiety symptoms, with a standard mean effect size of −0.41 (*p* < 0.0001) [22]. Herbal medicine, Gamisoyo-San, for 8 weeks showed a significant 12% increase (*p* = 0.021) in the QoL (WHOQOL-BREF scores) compared with the placebo-treated control group in patients diagnosed with GAD in Korea [21]. The study, which reviewed 15 studies on Chuna therapy and anxiety disorders, found that 10 randomized controlled trials (RCTs) showed an improvement compared to pretreatment levels, and three case reports indicated a therapeutic efficacy of over 90% [31]. However, KM typically refers to treatment care that provides and applies several types of treatment, such as acupuncture and herbal medicines, together rather than providing only a single therapy in the field.

Therefore, to understand the actual value of KM for anxiety disorders, the outcomes need to be examined in on-site practice rather than in an experimental setting that compares treatment arms confined to single or dual remedies. Two herbal medicines, Gami-ondam-tang and Gami-guibi-tang, were provided and incorporated in the 4-week KM treatment course in the present study. While the effectiveness was not driven only by the contribution of herbal medicine, the percentage improvement of QoL (8.4% (*p* < 0.001) on EQ-5D and 12% (*p* < 0.001) on EQ-VAS) was similar to the results for Gamisoyo-San in the aforementioned study [21]. However, little research has been carried out to reach a consensus on the effectiveness of KM. For herbal medicine, there is a consensus on the ingredients used for mental disorders such as anxiety [32,33]. However, a review of the effects of acupuncture on anxiety disorders found that the quality of the literature was poor, and the inconsistency of acupuncture points made it difficult to establish its effectiveness [34]. Therefore, more research is needed in this area.

Recent research has recognized the effectiveness of holistic approaches in the treatment of mental disorders [35,36,37]. Holistic therapy considers a patient’s physical, mental, and spiritual health as a whole rather than focusing on their symptoms. This type of medicine combines natural and alternative therapies to provide powerful healing benefits with fewer side effects compared with conventional treatment [38]. Holistic approaches have been shown to reduce anxiety and depressive symptoms in adolescents and mothers, who are more likely to experience anxiety disorders [35,36]. Therefore, holistic therapy is thought to effectively alter several anxiety-related indicators, such as QoL, not only in patients with mild symptoms but also in those with sociopsychological vulnerabilities [39]. Furthermore, the Korean Medicine Clinical Practice Guidelines recommend conducting screening tests upon the first visits of patients with anxiety disorders and providing KM treatments utilizing herbal medicine, acupuncture, and KM psychotherapy based on the patient’s pattern or disease progression [40]. KM offers a holistic approach to health and well-being, recognizing the importance of balance and harmony in maintaining well-being rather than viewing the body as a collection of separate parts to be treated individually [41,42]. By integrating the various modalities of herbal medicine, acupuncture, aromatherapy, and psychotherapy, it is possible to holistically improve the patient’s QoL.

Our research revealed that the KM treatment of anxiety resulted in improvements not only in anxiety symptoms but also in those of anger and depression. Anxiety disorders generally precede the onset of depressive disorders, and anxiety and depression often develop together [43,44]. Changes in neural circuits—specifically in the prefrontal-limbic pathways that govern emotional regulation—are a shared feature of both anxiety and depressive disorders [45]. Anger and anxiety, both stress-related emotions, often coexist due to shared psychological mechanisms, such as intolerance of uncertainty or anxiety sensitivity and reactions to perceived threats [46]. As reported in the Netherlands Study of Depression and Anxiety, 67% of patients with a primary diagnosis of depression also had concurrent anxiety disorders. Similarly, 63% of patients with primary anxiety disorder are also affected by depression [47]. As approximately one-third (29–32%) of patients with anxiety disorders experience anger attacks [48], not only are anxiety, anger, and depression inter-related, but they also influence QoL and life satisfaction [16,18,49]. Therefore, when measuring the effect of anxiety treatment, it is crucial to evaluate these associated conditions of anxiety. SSRI and SNRI, commonly used as first-line treatment for anxiety disorders, are known to reduce anxiety as well as alleviate symptoms of depression and anger [50]. Sertraline, in particular, has been reported to have a positive effect on anger control in patients with depression [51]. Although a direct comparison of efficacy is difficult, our findings appear to align with these findings in WM. In our study, a significant improvement in the clinical symptoms common to anxiety and depressive disorders was observed, which showed that KM treatment was generally effective in addressing mental health conditions in patients. This improvement also has an overall impact on QoL, which can be interpreted as a positive factor for economic evaluation indicators. However, when we evaluated changes in anger expression (AXI-K), we found that anger suppression (AXI-K-I) and anger expression (AXI-K-O) decreased after 4 weeks of treatment, while anger control (AXI-K-C) did not change significantly between the beginning and end of the treatment. This suggests that the treatment had a positive effect on anger suppression and anger expression, which are the simpler and emotion-driven aspects of anger, but the treatment did not yield a significant change in anger control, which is a more complex aspect that involves social skills. This treatment process may have limitations that prevent it from fundamentally addressing anger arising from the overall difficulty of living with anxiety. In addition, the correlation with QoL indicators was higher for anger than for anxiety, suggesting that anger acts as an extended secondary emotional problem of anxiety and has a negative impact on one’s overall life.

While KM has been considered to bring greater clinical and holistic benefits, especially for individuals with mental illness, its usage remains relatively low in Korea. In our comparison of the utilization of KM and WM in patients diagnosed with anxiety from the 2013 National Health Insurance Service sample cohort of Korea, of the 7811 patients with anxiety, only 2% (n = 151) visited KM clinics. The limited use of KM for anxiety care could be attributed to two main factors. First, KM clinics specializing in neuropsychiatry are not prevalent; in 2022, only 24 out of 15,095 KM clinics in Korea specialized in KM neuropsychiatry. Moreover, the number of active KM medicine doctors with a board-certified specialty in neuropsychiatry (with the title KMNPS) is relatively low (n = 300, about 1.25% of KM doctors) [52]; additionally, several types of KM treatment, such as herbal medicine or aromatherapy, are not covered by the National Health Insurance Service and, hence, they need to be paid for by the patients. In this study, the average out-of-pocket cost was 76% of the total cost over the course of a 4-week treatment, which could prevent low-income or elderly patients from accessing KM services.

Despite its significance, our study has some limitations. Our study is an observational study conducted in two KM clinics, so the number of participants is relatively low. However, the KM clinic where the study was conducted has the advantage of having KM doctors who are board-certified in neuropsychiatry, which ensures high-quality treatment. Although it may be overestimated to generalize to all KM clinics, the estimates from real-world data are meaningful in terms of reflecting actual clinical practice. Second, the treatment regimen given in this study was not precisely defined due to variations in the regimen (duration of treatment, type of herbal medicine, etc.) across patients. As mentioned, KM was transferred to each patient in a customized and personalized regimen according to patient symptoms. Thus, it is difficult to formulate a universal treatment regimen that could be offered to any patient with an anxiety disorder in the context of KM. Future comparative studies with the antidepressants and anti-anxiety medications traditionally used to treat anxiety disorders may complete our understanding of the effectiveness of KM. More research is needed to prove the effectiveness of KM on anxiety disorders.

## 5. Conclusions

Our retrospective observational study suggested that KM, which includes acupuncture, herbal medicine, and psychotherapy, can effectively treat patients with anxiety disorders, leading to significant improvements in both clinical symptoms and QoL. Our results showed that addressing anxiety not only alleviated anxiety symptoms but also showed beneficial effects on the symptoms of anger and depression. Furthermore, by presenting our cost outcomes, we have contributed valuable data to future cost analyses in KM research. In light of the findings, policymakers may wish to expand insurance benefits to include KM therapies, such as herbal medicine or aromatherapy, to improve the efficiency of treatment and health outcomes in patients with anxiety disorders.

## Figures and Tables

**Figure 1 healthcare-12-01345-f001:**
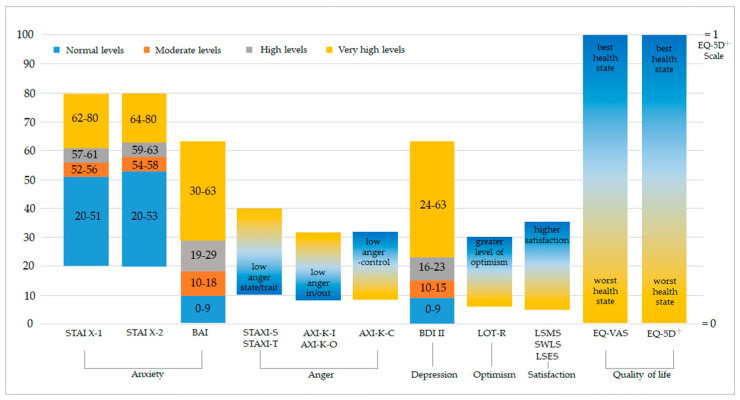
Psychological and health-related quality of life measures.

**Figure 2 healthcare-12-01345-f002:**
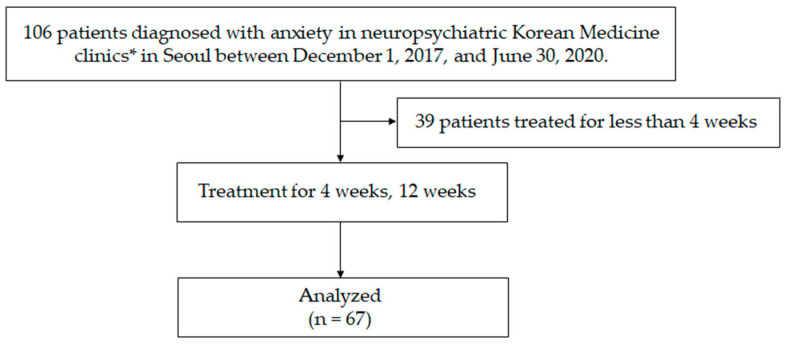
Flowchart of the study. * Among the patients who visited two clinics in Seoul for treatment, 96 were recruited from one clinic and 10 from the other clinic.

**Figure 3 healthcare-12-01345-f003:**
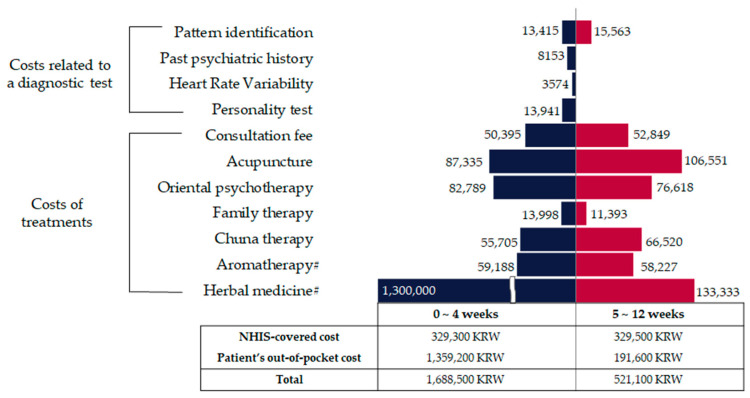
Average costs of Korean medicine treatments for patients with anxiety disorders over 4 and 12 weeks (KRW). The diagnostic cost is the one-time expense incurred during the initial visit, while each of the treatment cost items represents the average cost over a 4-week or 8-week period. USD 1 = KRW 1086.3 (2020). National Health Insurance Service: NHIS. # Herbal medicine and aromatherapy were not covered by the NHIS.

**Table 1 healthcare-12-01345-t001:** Demographic data for the patients with anxiety disorder (n = 67).

Age (mean ± SD)	33.2 ± 10.7
Type (%)	
GAD	30 (44.8)
PD	26 (38.8)
PTSD	2 (3.0)
SAD	9 (13.4)

GAD: generalized anxiety disorder; PD: panic disorder; PTSD: posttraumatic stress disorder; SAD: social anxiety disorder; SD: standard deviation.

**Table 2 healthcare-12-01345-t002:** Changes in clinical outcomes over 4 weeks in patients with anxiety disorder (n = 67).

	Range	Baseline Mean (SD)	After 4 Weeks Mean (SD)	Improvement from Baseline (%)	*p*-Value
Anxiety					
STAI X-1 #	20–80	59.24 (9.55)	50.48 (12.67)	14.6	<0.001 ***
STAI X-2 #	20–80	57.33 (9.98)	51.57 (11.77)	9.6	<0.001 ***
BAI #	0–63	28.15 (12.74)	18.72 (13.47)	22.5	<0.001 ***
Anger					
STAXI-S #	10–40	16.39 (7.28)	15.33 (7.43)	3.5	0.135
STAXI-T #	10–40	20.30 (7.22)	19.48 (6.79)	2.7	0.237
AXI-K-I #	8–32	18.22 (5.29)	17.01 (5.28)	5.0	0.012 ***
AXI-K-O #	8–32	14.31 (5.17)	13.58 (5.48)	3.0	0.047 ***
AXI-K-C	8–32	21.70 (5.12)	21.61 (4.78)	−0.4	0.817
Depression					
BDI II #	0–63	25.78 (12.04)	18.6 (13.8)	11.4	<0.001 ***
Optimism					
LOT-R	6–30	11.81 (4.04)	12.63 (3.70)	3.4	0.016 ***
Satisfaction					
LSMS	5–35	30.24 (3.97)	30.25 (4.20)	0.1	0.975
SWLS	5–35	15.31 (5.89)	15.58 (5.72)	0.9	0.556
LSES	5–35	21.63 (7.15)	23.48 (7.32)	6.2	0.003 ***
Quality of life					
EQ-5D	0–1	0.75 (0.17)	0.83 (0.14)	8.4	<0.001 ***
EQ-VAS	0–100	49.54 (22.96)	61.39 (21.11)	11.9	<0.001 ***

AXI-K-C: Anger Expression Inventory Anger-Control; AXI-K-I: Anger Expression Inventory Anger-In; AXI-K-O: Anger Expression Inventory Anger-Out; BAI: Beck Anxiety Inventory; BDI II: Beck Depression Inventory-II; EQ-VAS: EuroQoL Visual Analog Scale; EQ-5D: Euro Quality of Life 5 Dimensions utility score; LOT-R: The Revised Life Orientation Test; LSES: The Life Satisfaction Expectancy Scale; LSMS: The Life Satisfaction Motivation Scale; SD: standard deviation; STAI: State-Trait Anxiety Inventory: axis1 (X-1, State), axis2 (X-2, Trait); STAXI-S: State-Trait Anger Expression Inventory State Anger; STAXI-T: State-Trait Anger Expression Inventory Trait Anger; SWLS: The Satisfaction with Life Scale. # A lower score indicates a higher degree of symptoms. *** *p* < 0.05.

## Data Availability

The data that support the findings of this study are available within the article.

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
