# Peer review of "The Mental Health Outcomes and Cost Estimates of Korean Medicine for Anxiety Disorder Patients"

_healthcare, 2024, doi:10.3390/healthcare12131345_

Round 1
Reviewer 1 Report
Comments and Suggestions for Authors
Dear Authors,
I have reviewed your manuscript titled "The Effectiveness and Cost Outcomes of Korean Medicine for 2 Anxiety Disorders: A Retrospective Case Series Study." Overall, I find your study to be valuable and well-organized. However, there are several areas that require further attention and revision.
· I believe the conclusions drawn from the study are overly ambitious. It would be beneficial to reassess whether your findings support such definitive statements.
· Using a term more frequently found in the literature instead of "Korean medicine" would improve readability and clarity for the readers.
· The abstract is complex and difficult to understand. It contains too many abbreviations and details, making the main takeaway unclear. Simplifying the abstract would enhance its comprehensibility.
· The introduction is well-summarized and purposefully structured.
· The methods section lacks information on how the economic analysis was conducted and what the evaluation criteria were. Providing more details in this area is necessary.
· Detailed demographic data of the patients should be included in the main text rather than in the supplementary file.
· I did not understand how the correlation table contributes to your hypothesis. Either provide more explanation or consider removing the table.
· I would prefer to see the flowchart, Table S4, and Figure S1 currently in the supplementary file included in the main text. This recommendation is based on the observation that readers often do not refer to supplementary files sufficiently.
I have also made some annotations on the attached document. I would appreciate it if you could take them into consideration.
Sincerely,

Reviewer 2 Report
Comments and Suggestions for Authors
The authors conducted a study to evaluate the effects of KM on anxiety disorders. There are some major and minor points for authors to consider to further improve the manuscript.
Major points
- Given that the authors aimed to evaluate the effectiveness and cost-effectiveness of KM, they should conduct a comparative study. They should have a control group, which I think would be non-KM-treated patients. Thus, this single-arm study evaluating pre- and post-treatment changes is subject to biases and should be interpreted cautiously.
- ACER is not an appropriate measure in an economic evaluation study. The authors should consider calculating the incremental cost-effectiveness ratio (ICER) between KM-treated patients and non-KM-treated patients. They could also consider presenting the findings separately as cost outcomes and QOL outcomes, but not ACER.
- Line 125-127: QALY should be calculated only in the observed period of 12 weeks.
- Line 69: “Overall effectiveness of KM”. Patients require different KM approaches according to their medical history and preferences. Thus, the treatments are too different from being pooled together as “KM.” I think the authors should analyze the effects of individual KM approaches and their combinations to demonstrate which approach or combination works.
Minor points
Title
- “Case series study” With 67 patients, I would not call it a case series study.
Introduction
- Paragraph 1: Consider adding background information on anxiety disorders in Korea to show the significance of the problem.
- Line 45: “WM”. Spell out the first usage of the abbreviation
Methods
- Were patients treated with anti-anxiety drugs?
- What was the study’s perspective?
- Line 92: Why did eligible patients need to receive treatment for more than 4 weeks?
- Line 129: ACER is not an appropriate measure in an economic evaluation study.
- Line 136: Did the authors mean “paired t-test”?
Results
- Line 153-154 and Table S3: What were the rationales for comparing gender results?
- Figure 2: Consider adding average total costs into the figure.
Comments on the Quality of English Language
The English language is generally good. I detect no issues except the abbreviations which might not be spelled out at first usage.
Reviewer 3 Report
Comments and Suggestions for Authors
Studies on the effectiveness of treatments of depression are clear important as it is an illness that regretfully has a relatively high prevalence. Economic considerations are also important. I have a few comments.
Some comments
Numerical measuring anxiety levels is a challenging teak. It is good that the authors added a section between anxiety-relevant measurements scales. However, this requires a bit more explanation and description of these measures as it is at the core of the paper.
I think that it should be mentioning that one of the limiting factors of the study is the relatively low number of patients 67 actually included in the analysis.
A related important point. In the sentence: “Of the 106 patients who visited the NKM clinics during the observation period, 67 148 who were treated for 4 weeks or longer were considered for analysis”… I understand that the ones excluded were not treated for four weeks or longer. I think this point needs to be make clearer. If the exclusion was due to other considerations this needs to be mentioned.
The conclusions section is too short. Either expand it or merger it with the discussion section.
I think that it is also important to support some of the treatments with more existing literature as on some cases there is no consensus view among researchers of the effectiveness of such treatments. As some of the techniques are not generally accepted in the academic community there needs to be more literature supporting their use. I am taking an agnostic position on them.
Minor comments
I think that there is some missing in the last page related to a disclaimer
English is overall ok but authors should consider reading it again and polishing it a bit.
Comments on the Quality of English LanguageEnglish is overall ok but authors should consider reading it again and polishing it a bit.
Round 2
Reviewer 2 Report
Comments and Suggestions for Authors
I am satisfied with the revision. I have only two suggestions.
- Figure 3: Showing which costs were covered by insurance and which were paid out-of-pocket by the patients would provide an insightful information to the audiences.
- Perspective should be clearly stated in the abstract and the methods sections.
Comments on the Quality of English Language- I detect few grammartical mistake. Please proofread the article once more.
